# ROYAL SOCIETY
# OPEN SCIENCE

# Research

materials science

suction casting, Al-Cu-Fe-Ce alloy, quasicrystal, thermal expansion properties, microstructure

**Author for correspondence:**
Zhong Yang
e-mail: yz750925@163.com

This article has been edited by the Royal Society of Chemistry, including the commissioning, peer review process and editorial aspects up to the point of acceptance.

# Influence of cerium content on the microstructure and thermal expansion properties of suction cast Al-Cu-Fe alloys

Juan Wang, Zhong Yang, Zhijun Ma, Yaping Bai, Hongbo Duan, Dong Tao, Guoqing Shi, Jiachen Zhang, Zhen Wang and Jianping Li

School of Materials and Chemical Engineering, Xi'an Technological University, Xi'an 710021, People's Republic of China

JW, 0000-0002-1019-8482; ZY, 0000-0001-9544-3450

Low-expansion alloys are of great importance and can be used for the development of new aerospace materials. Herein, we report diverse rare earth quasicrystal alloys fabricated by the vacuum suction casting process. The effects of the addition of cerium (Ce) on the microstructure, thermal expansion properties and microhardness of the Al-Cu-Fe alloy were systematically investigated. This study discovered the tiny Al-Cu-Fe-Ce microstructure. A uniform distribution could be achieved after Ce addition amount is elevated. At the Ce addition amount of 1 at%, the lowest alloy thermal expansion coefficient was obtained. The alloy exhibited the maximum microhardness under these conditions. The microhardness of alloys containing 1 at% of Ce was approximately 2.4 times higher than the microhardness exhibited by alloys devoid of Ce additives. The coefficient of thermal expansion decreases by approximately 20%. The use of the suction casting process and the addition of an appropriate amount of Ce can potentially help design and develop Al-Cu-Fe-Ce alloys.

# 1. Introduction

Three decades ago, quasicrystal phase was originally detected in Al-Mn alloys under rapid solidification [1]. The property of high room temperature brittleness limits the wide application of quasicrystal alloys. However, quasicrystals have integrated characteristics, like great hardness [2], elastic moduli and strength. The materials also exhibit low surface energy, favourable wear/corrosion resistance and low coefficient of friction [3,4]. Thus, they can be used as the reinforcing phase of

soft matrix composites [5–9]. The wettability and bonding force between the quasicrystal and matrix in reinforced soft matrix composites are better than those observed in distributed ceramic particles (such as SiC and AlN) [10]. These impart excellent strength and bearing capacity [11,12] to the composites. The quasicrystalline particles exhibit excellent thermal expansion properties [13]. When they are associated with the matrix, composites exhibiting stable size and excellent high-temperature mechanical properties can be obtained [14,15].

Among the Al-systems with the potential to form I-phase, Al-Cu-Fe alloys have attracted much attention because of the insufficient toxicity due to their facile manufacturing and cost-effectiveness in terms of the alloying elements. They also exhibit high thermal stability [16–20]. However, the formation range of the I-phase is narrow. The quasicrystal alloys prepared by the conventional casting process contain less numbers of quasicrystals and exhibit complex phases and more casting defects (such as porosity, shrinkage cavity, etc.). In most cases, the as-cast quasicrystal alloy is annealed [21,22] to increase the quasicrystal content. However, the annealed alloy contains a lot of casting defects, which significantly influence the performance of the materials.

At present, people often follow the process of alloying for the sake of improving material properties and structure, especially those of rare earth elements. Rare earth elements, important mineral resources, are considered as industrial vitamins. They exhibit special electronic structure and chemical activity. These are considered as suitable microalloying elements. Adding rare earth affects alloy characteristics and microstructure in the following aspect: they form stable rare earth compounds that exhibit high melting points. Enhanced strength and thermal stability of the alloys were achieved. In addition to serving as the main reinforcing phase in conventional alloys, the elements play a strengthening role in other amorphous alloys in the form of oxides. They can be used as grain refiners in aluminium melts. They can also be used as de-gasifying and de-slagging agents as they react with gas and liquid impurities [23].

Many studies on the rare earth microalloying process have been reported. For instance, some study has examined the impact of adding trace cerium (Ce) and Y on Mg–Zn–Zr alloy sheet mechanical performance and microstructure [24]. According to their observations, adding Ce and Y significantly refined the grains. The tensile strength could be significantly increased. Fang *et al*. [25] examined how different Ce/La misch metal addition amounts affected Mg-Zn-Y alloy characteristics and microstructure. The results showed that the high-temperature mechanical properties of Mg-Zn-Y alloy were significantly improved when 1 wt% of the Ce/La was added. Zhang *et al*. [26] reported that the addition of an appropriate volume of rare earth simplified aluminium melt purification. The process can enhance the strength of the alloys. As for Al-3.0 wt% Mg alloys, their electrical resistivity can be downregulated. Guo *et al*. [27] studied how Er addition into Al-Fe-Si alloy affected microstructure, phase composition, mechanical performance and electrical conductivity. According to their findings, during melting, Er (a rare earth element) removed those impurity elements Ti, V and Cr within the alloy. Besides, adding Er remarkably enhanced alloy characteristics and microstructure.

However, the production cost increases after the addition of excessive rare earth element. On the basis of Al-Cu-Fe alloy, this study fabricated diverse novel Al-Cu-Fe-Ce quasicrystal alloys from suction casting. Different amounts of Ce were used as additives. This study explored the mechanism by which Ce affected alloy thermal expansion behaviour, hardness and microstructure. The results obtained from the study of the novel quaternary rare earth quasicrystal alloy can help develop quasicrystal materials.

# 2. Experimental details

In this study, the following raw materials were used, including high-purity Al (99.99%), Cu, Al-30Ce (99.5%, wt%) and Al-60Fe (99.98%) master alloys. In brief, each raw material was placed in the copper crucible within the vacuum chamber filled with argon for preventing oxidation. Afterwards, the arc was used to melt the materials, followed by the flowing of cold water surrounding copper crucible walls to achieve rapid cooling to ambient temperature in 5 min. Thereafter, the resultant sample was subjected to flipping and re-melting four times for improving uniformity.

Then, by vacuum suction casting, we acquired cylindrical samples whose lengths and diameters were 15 and 2 mm, respectively. The ingot cooling rate was between approximately $10^2$ K s$^{-1}$ (edge) and approximately $10^3$ K s$^{-1}$ (centre).

Metallographic samples were prepared following the cold insertion method. The specimens were mechanically polished using diamond sandpapers. Microstructures were analysed using scanning electron microscopy (SEM, Tescan, VEGA II-XMU). The SEM instrument was coupled with high-resolution

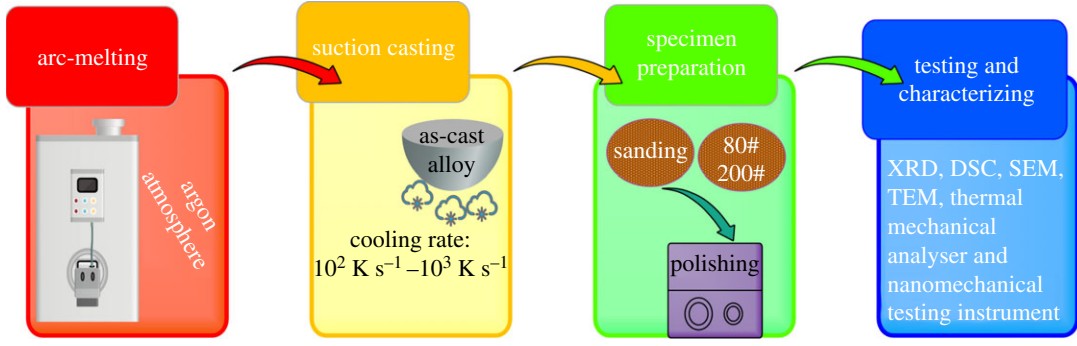

**Figure 1.** Processing procedure for the preparation and observation of quasicrystal alloys.

transmission electron microscopy (HRTEM, JM-2010) as well as energy-dispersive X-ray spectroscopy (EDS). To conduct SEM analysis, we adopted the diluted Keller's reagent (consisting of 2.5 ml $HNO_3$, 1.0 ml HF, 1.5 ml HCl and 95 ml water) to be the etchant for those samples polished. We also conducted EDS point analysis and X-ray elemental mapping for exploring diverse phase nature and elemental distribution of Al-Cu-Fe-Ce samples.

In the presence of Cu Kα radiation (λ = 0.1542 nm), we recorded $(Al_{63}Cu_{25}Fe_{12})_{100-x}Ce_x$ (x = 0, 0.1, 0.5, 1 and 2 at%) alloys for their X-ray diffraction (XRD-6000) patterns. The scanning speed and scanning angle were set at $4°$ $min^{-1}$ and $20°–90°$, respectively. Meanwhile, thermal analysis was conducted using high-temperature differential scanning calorimetry (TGA, DSC1). The experiments were conducted under an atmosphere of Ar, and we set the heating rate at $20°C$ $min^{-1}$. The mass of the sample used for the DSC experiments was approximately 30 mg. We applied Image-Pro Plus 6.0 software in quantitatively analysing SEM images. Microhardness tests of the alloys were conducted using a Bruker Hysitron TI Premier nanomechanical testing instrument operated at a temperature of $25°C$. We prepared $(Al_{63}Cu_{25}Fe_{12})_{100-x}Ce_x$ alloy into the dimension of $\varphi 2$ mm × 5 mm for investigating its thermal expansion characteristics. This experiment adopted the thermal and mechanical analyser (TMA, SDTA-840). In the experimental process, we heated the specimen between $30°C$ and $500°C$ at the $5°C$ $min^{-1}$ heating rate in Ar atmosphere The computer was used to record all data, while Mettler Toledo thermal analysis system was adopted for data analysis. The data obtained after the system test were used to calculate the average thermal expansion coefficients of samples with different Ce contents at $30–100°C$, $30–200°C$, $30–300°C$, $30–400°C$ and $30–500°C$ by Excel software. The experimental process is shown in figure 1.

# 3. Results

## 3.1. X-ray diffraction analysis

The results obtained by analysing of the XRD images of $(Al_{63}Cu_{25}Fe_{12})_{100-x}Ce_x$ (x = 0, 0.1, 0.5, 1 and 2 at%) are shown in figure 2. The phases in the $Al_{63}Cu_{25}Fe_{12}$ alloy primarily consisted of the I-phase, β-$Al_{0.5}Fe_{0.5}$ phase (β-phase) and θ-$Al_2Cu_3$ phase (θ-phase). When Ce was added, the most prominent peaks observed in the image of the Al-Cu-Fe-Ce (x = 0.1, 0.5, 1 and 2 at%) alloys corresponded to the I-phase, β-phase and $Al_{13}Ce_2Cu_{13}$ phase. The results indicated that the peak intensity of the β-phase decreased, the intensity of the diffraction peaks of the $Al_{13}Ce_2Cu_{13}$ phase increased and the peak intensity of the I-phase increased as the Ce content in the alloy was increased. When the content of Ce was 1 at%, the peak intensity corresponding to the I-phase was the maximum, and the peak intensity corresponding to the β-phase was the minimum.

## 3.2. Differential scanning calorimetry analysis

Figure 3 presents DSC curves of $(Al_{63}Cu_{25}Fe_{12})_{100-x}Ce_x$ (x = 0, 0.1, 0.5, 1 and 2 at%) alloys prepared through suction casting. The curve corresponding to the Al–Cu–Fe alloy (x = 0) revealed the presence of the θ, I and β-phases, when the melting temperatures were $681.8°C$, $876.2°C$ and $968°C$, respectively. As revealed by the curves corresponding to Al–Cu–Fe–Ce samples after suction casting, there were three phases, including $Al_{13}Ce_2Cu_{13}$ (at a $687°C$ melting temperature), I and β-phases.

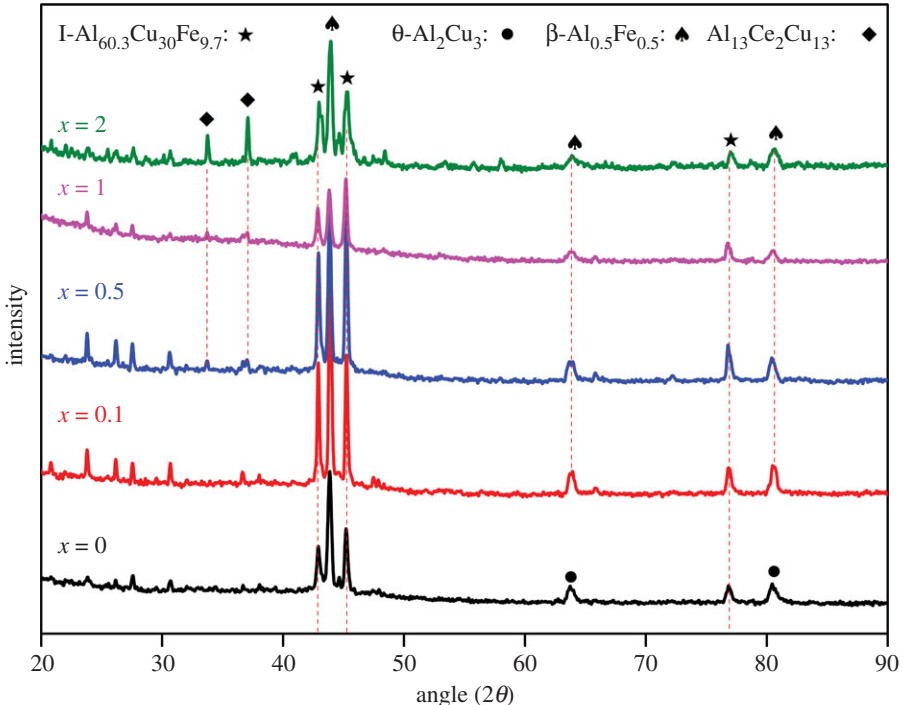

**Figure 2.** XRD patterns obtained following suction casting of $(Al_{63}Cu_{25}Fe_{12})_{100-x}Ce_x$ ($x = 0$, 0.1, 0.5, 1 and 2 at%) samples.

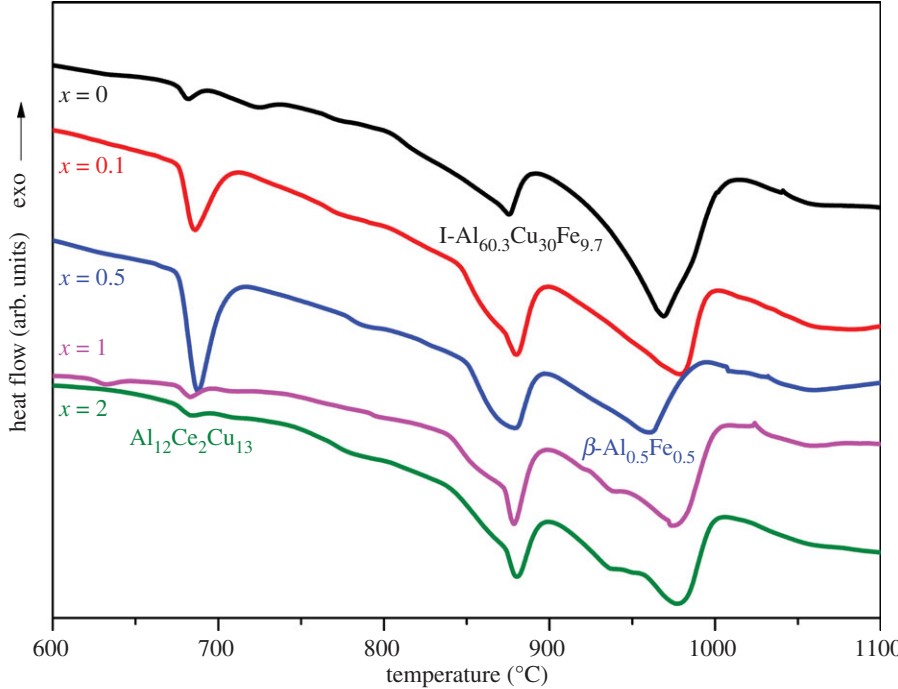

**Figure 3.** DSC curves of suction cast $(Al_{63}Cu_{25}Fe_{12})_{100-x}Ce_x$ ($x = 0$, 0.1, 0.5, 1 and 2 at%) samples.

These results agree well with the results obtained from the XRD experiments. The I-phase transformation temperature was slightly different between Al–Cu–Fe–Ce and Al–Cu–Fe alloys. Such finding was associated with the effect of undercooling effect on samples prepared by suction casting. The notable kinetic undercooling property achieved due to the extraordinary cooling on the molten metallic melt was the critical property observed during the suction casting process. Non-equilibrium solidification effects were also observed. The presence of Ce results in an increased degree of undercooling at liquid–solid interface in solidification. Such increased undercooling is associated with facile I-phase nucleation.

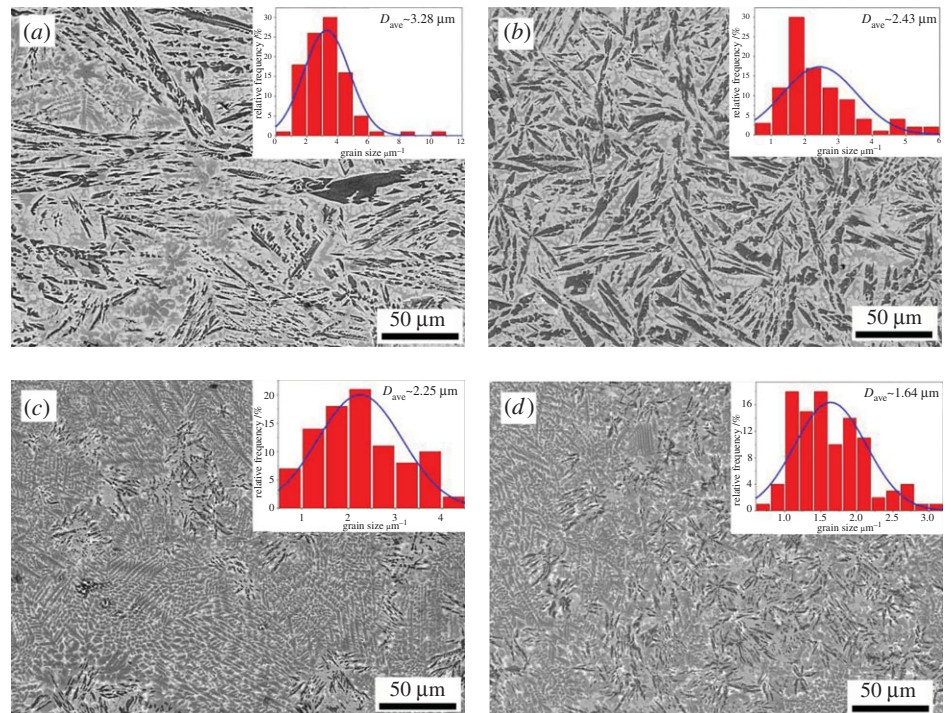

**Figure 4.** Microstructures of suction cast $(Al_{63}Cu_{25}Fe_{12})_{100-x}Ce_x$ ($x = 0$, 0.1, 1 and 2 at%) samples and the size distribution histograms of the dark grey areas (inset: $D_{ave}$ denotes the average grain size of the dark grey areas, where (a) $x = 0$, (b) $x = 0.1$, (c) $x = 1$ and (d) $x = 2$.

## 3.3. Scanning electron microscopy and X-ray spectroscopy analysis

Figure 4 presents the SEM images for $(Al_{63}Cu_{25}Fe_{12})_{100-x}Ce_x$ ($x = 0$, 0.1, 1 and 2 at%) samples. The microstructure of the alloy in the absence of Ce ($x = 0$) was relatively coarse. Black dendritic and long strip phases were observed. When 0.1 at% of Ce was added, the microstructure of the alloy did not vary significantly. However, the distribution of the dendritic structure was fine, and it was uniformly distributed. The dendritic phase was reduced, and the microstructure presented a uniform strip distribution. When 1 at% of Ce was added, the alloy exhibited a faint needle dark grey phase. A small amount of the fine dendritic and white phases were observed. As the content of Ce was increased, the grain refinement effects significantly increased. The grain refinement of the dark grey areas of the alloys increased.

The size distribution histogram of the dark grey phases of the alloys is presented in the corresponding insets in figure 4a–d. The histograms were obtained by analysing the statistical data using Image-Pro plus 6.0 software. When the Ce content was increased from 0 to 2 at%, Cu-Fe-Ce alloy grain size significantly reduced from 3.28 to 1.64 μm in dark grey phase.

EDS experiments were conducted for the representative areas, as shown in figures 5 and 6, to identify the nature of the phases present in the $(Al_{63}Cu_{25}Fe_{12})_{100-x}Ce_x$ alloy. It was observed that the $Al_{63}Cu_{25}Fe_{12}$ alloy consisted of three constituent areas: black, dark grey and light grey, and every area showed distinct concentration (figures 4 and 5). Based on these results and the results obtained from XRD studies, we observed that the above three areas stood for three distinct phases, namely, β, I and θ-phases. As observed from the SEM image, there was 6.98 at% iron in the light grey area, which stood for θ-phase. Meanwhile, there was a low copper content in the dark grey area (9.93 at%, as observed on EDS), which stood for β-phase, whereas the rest of the grey area represented I-phase [28].

Figure 4b–d shows the representative micrograph of the Al-Cu-Fe-Ce alloy. As the amount of Ce was increased, the microstructure of the alloy varied, but the phases of the Al-Cu-Fe-Ce alloy remained the same. The $(Al_{63}Cu_{25}Fe_{12})_{99}Ce_1$ alloy was used as an example to identify the phases. It consisted of three different colours: dark grey, grey and white. In the white area, Al, Ce and Cu were present in the atomic ratio of $13:2:13$ (figure 6). One can confirm that the white area in figure 2 corresponded to the $Al_{13}Ce_2Cu_{13}$ phase. Meanwhile, the black areas and the black-grey areas represented the β-phase and the I-phase, respectively.

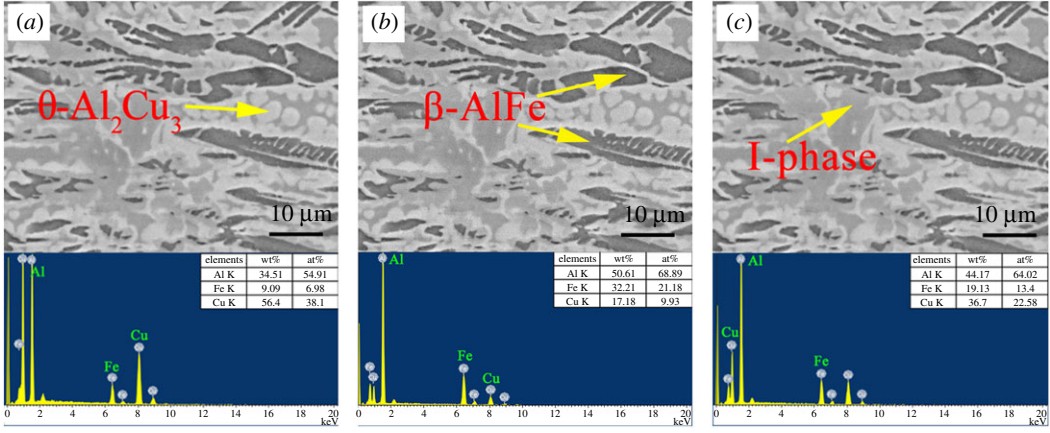

**Figure 5.** Microstructures of suction cast $Al_{63}Cu_{25}Fe_{12}$ alloy: EDS data of (a) θ-phase, (b) β-phase and (c) I-phase.

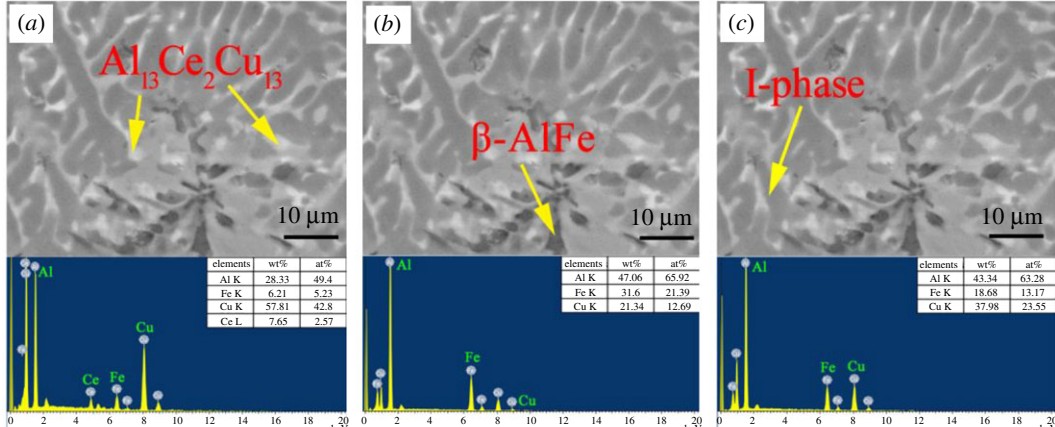

**Figure 6.** Microstructures of suction cast $(Al_{63}Cu_{25}Fe_{12})_{99}Ce_1$ alloy: EDS data of (a) $Al_{13}Ce_2Cu_{13}$ phase, (b) β-phase and (c) I-phase.

Analysis of figures 5 and 6 revealed that when 1 at% of Ce was added, the proportion of each element in I-phase showed a slight change compared with that of $Al_{63}Cu_{25}Fe_{12}$ alloy, because of the great cooling rate during the suction casting process and the insufficient diffusion of the alloying elements.

We determined the area fractions of each phase in the suction cast $(Al_{63}Cu_{25}Fe_{12})_{100-x}Ce_x$ ($x = 0.1$, 0.5, 1 and 2 at%) alloys by adopting Image-Pro Plus 6.0 software. The results of the calculation are shown in table 1. Analysis of the microstructures of the samples and the data presented in table 1 indicates that the addition of Ce results in increased formation ability of the I-phase. The area fraction of the β-phase decreased. The I-phase area fraction increased when 1% of Ce was added.

## 3.4. X-ray element mapping and transmission electron microscopy technique

As a result, $Al_{13}Ce_2Cu_{13}$ and I-phases showed enhanced production within the alloy, with progressively decreased β-phase production as the Ce content was increased. The $(Al_{63}Cu_{25}Fe_{12})_{99}Ce_1$ alloy containing the largest amount of the I-phase was subjected to analysis to determine the element distribution and the selected area electron diffraction (SAED) pattern of each phase.

Figure 7 displays the enlarged SEM images, corresponding EDS maps, and the corresponding SAED patterns of each phase in the $(Al_{63}Cu_{25}Fe_{12})_{99}Ce_1$ alloy. Figure 7b–e presents phase EDS maps within $(Al_{63}Cu_{25}Fe_{12})_{99}Ce_1$ alloy. It is found that Cu, Fe and Al contents increased in I-phase, with no Ce. The β-phase contained fewer amounts of copper. The $Al_{13}Ce_2Cu_{13}$ phase contained Al and Cu but was devoid of iron. Segregated Ce was present in the grain boundaries.

Figure 7f–h shows the corresponding SAED pattern analyses of each phase in the $(Al_{63}Cu_{25}Fe_{12})_{99}Ce_1$ alloy. Figure 7f reveals the SAED patterns of the I-phase. The fivefold rotational symmetry is a characteristic of the I-phase. The diffraction patterns further confirmed that the I-phase was preserved in the Al-Cu-Fe-Ce alloy. Figure 7g shows the SAED patterns recorded along with the [1 2 0] plane of the $Al_{13}Ce_2Cu_{13}$ phase. $Al_{13}Ce_2Cu_{13}$ belongs to Al–Cu–Ce system. It is cubic with $a = 1.189$ nm and

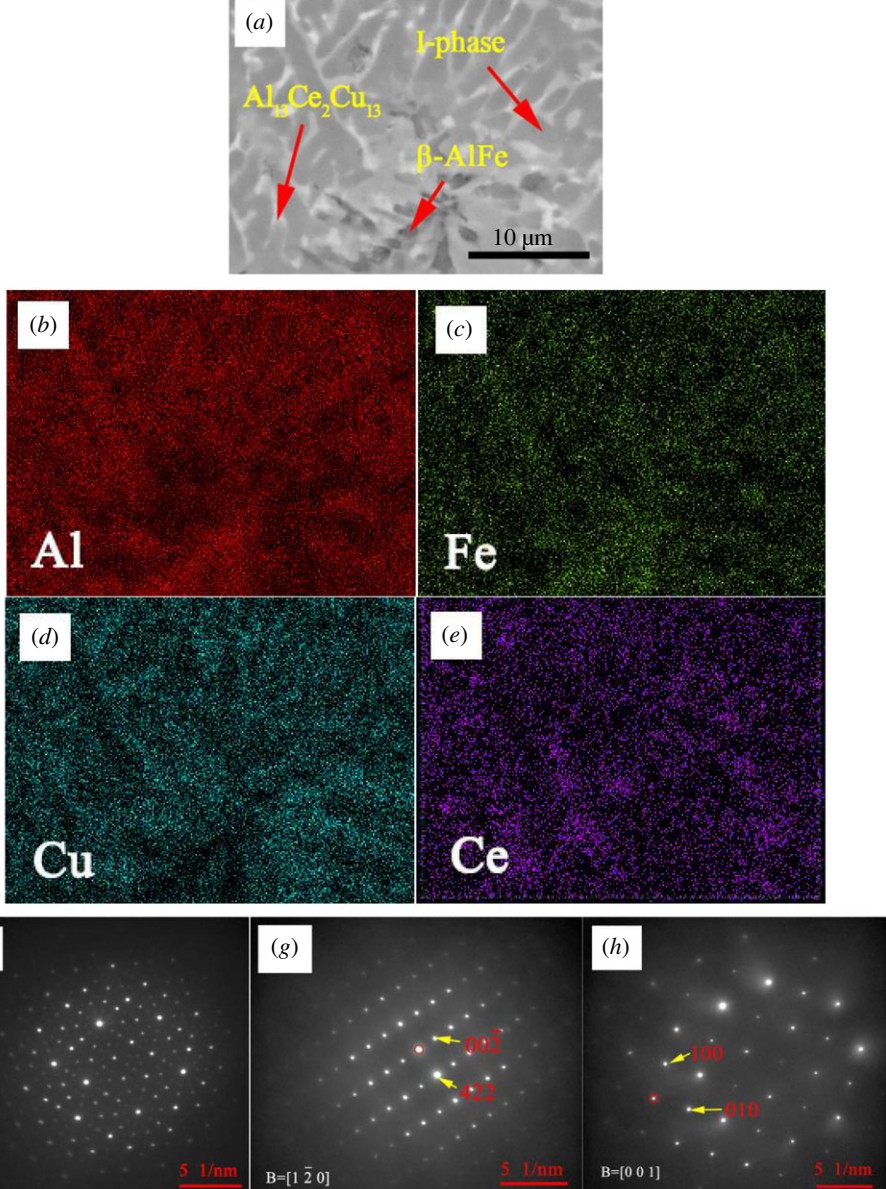

**Figure 7.** Microstructure of the $(Al_{63}Cu_{25}Fe_{12})_{99}Ce_1$ alloy: (*a*) SEM image, (*b–e*) EDS maps, (*f–h*) corresponding SAED patterns of the I-phase, $Al_{13}Ce_2Cu_{13}$ phase and β-phase in (*a*).

**Table 1.** Area fraction of each phase in the suction cast $(Al_{63}Cu_{25}Fe_{12})_{100-x}Ce_x$ ($x = 0.1$, 0.5, 1 and 2 at%) alloys.

| alloy | area fraction of each phase (%) | | |
|---|---|---|---|
| | I-phase | $Al_{13}Ce_2Cu_{13}$ | β-phase |
| $x = 0.1$ | 46.6 | 3.3 | 50.1 |
| $x = 0.5$ | 60.45 | 5.2 | 34.35 |
| $x = 1$ | 77.45 | 6.8 | 15.75 |
| $x = 2$ | 66.45 | 8.1 | 25.45 |

$\alpha = 90°$ (space group Fm-3c). Figure 7*h* shows the SAED patterns taken along with the [001] plane of the β-phase. It is cubic with $a = b = c = 0.291$ nm and $\alpha = \beta = \gamma = 90°$ (space group Pm-3m). The β-phase belongs to the quasicrystal approximate phase.

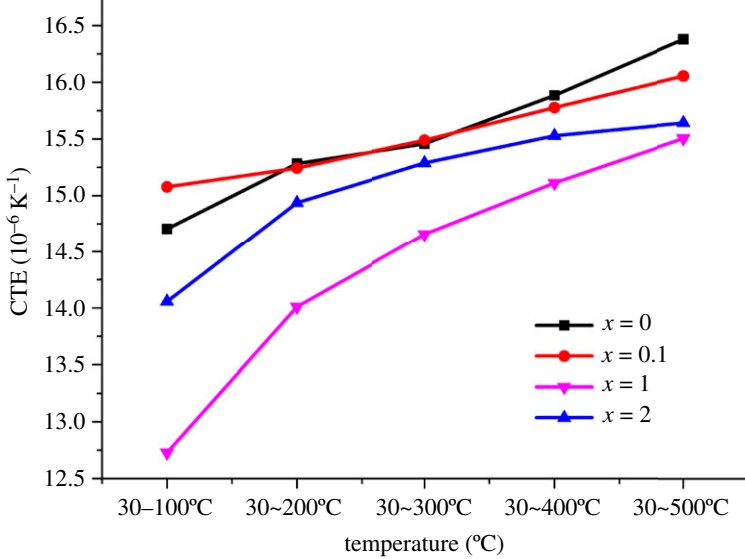

**Figure 8.** Effect of temperature on the linear expansion coefficients of suction cast $(Al_{63}Cu_{25}Fe_{12})_{100-x}Ce_x$ ($x = 0$, 0.1, 0.5 and 1 at%) samples.

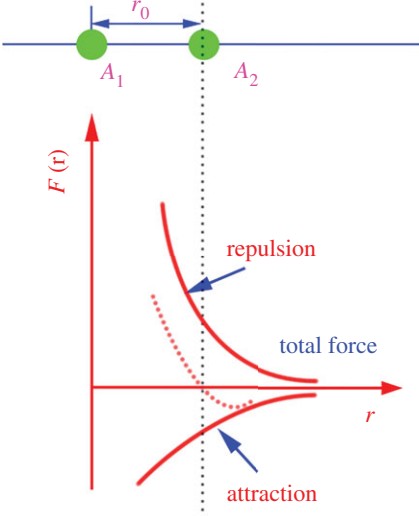

**Figure 9.** Diagram of the physical nature of thermal expansion.

## 3.5. Thermal expansion properties

Thermal expansion characteristics can be characterized by the average thermal expansion coefficient. For $(Al_{63}Cu_{25}Fe_{12})_{100-x}Ce_x$ ($x = 0$, 0.1, 0.5 and 1 at%) samples prepared by suction casting, their mean coefficient of thermal expansion (CTE) values vary with temperature (30°C –500°C), as shown in the figure 8.

The CTE of all the samples gradually increased with temperature (figure 8). At the same time, the longitudinal comparison showed that with the increase of Ce content, the thermal expansion coefficient of the alloy first decreased and then increased. When the content of Ce was 1 at%, the thermal expansion coefficient of the alloy at each temperature stage was the smallest. The CTE in the temperature range of 30–100°C was $14.71 \times 10^{-6}$ K$^{-1}$ for the sample devoid of Ce. When the Ce content was 1 at%, the CTE in the temperature range of 30–100°C was $12.73 \times 10^{-6}$ K$^{-1}$, which was approximately 20% lower than that exhibited by the alloy devoid of Ce. These observations can be attributed to the fact that the alloys are prepared following the same process, and the test conditions of the CTE were also the same. The results revealed that the content of Ce was the key to the CTE value.

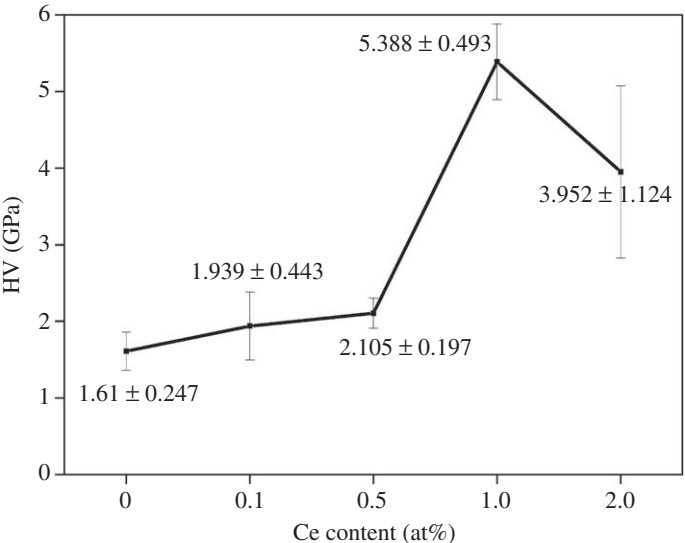

**Figure 10.** Microhardness of Al-Cu-Fe-Ce under varying Ce content.

**Table 2.** Mismatch between each solute element (%).

|    | Al   | Cu   | Fe   |
|----|------|------|------|
| Ce | 27.3 | 42.2 | 43.3 |

The CTE of solid materials can be described by the atomic model presented in figure 9. The corresponding formula (3.1) [29] is as follows:

$$\alpha = \frac{3kg}{R_0 m^2 \omega^4},$$  (3.1)

where

$R_o$—the average interatomic distance at a temperature of 0 K;

$k$—Boltzmann's constant;

$g$—the third-order differential of the potential function at position $R_0$;

$m$—the average mass of the atom;

$\omega$—the vibration frequency of the atom.

When the Ce content was increased, the sizes of the grains in the alloys became smaller and evenly distributed. A stronger blocking effect on dislocations was achieved. That is, the binding energy between the atoms was greater. According to formula (3.1), the greater the increase in potential energy of the atom (as it deviated from its average position), the greater the $R_o$ and the smaller the expansion coefficient (figure 9).

## 3.6. Microhardness

Figure 10 reveals the microhardness of the Al-Cu-Fe-Ce alloy containing varying amounts of Ce. The average hardness value of 1.61 GPa was obtained for the $Al_{63}Cu_{25}Fe_{12}$ alloy, and the average hardness values of 2.105, 3.952, 5.388 and 1.939 GPa were obtained for the $(Al_{63}Cu_{25}Fe_{12})_{100-x}Ce_x$ ($x$ = 0.1, 0.5, 1 and 2 at%, respectively) samples.

When the content of Ce ranged from 0 to 2 at%, the alloy microhardness value showed the first increase and later decrease trend. When Ce = 1 at%, the alloy exhibited the largest microhardness value. This can be attributed to the fact that the addition of Ce will cause constitutional overcooling, leading to grain refinement. Fine-grain strengthening was also achieved. The addition of Ce forms the $Al_{13}Ce_2Cu_{13}$ phase. Analysis of the microstructure of the alloy (figures 4 and 6) revealed that the white $Al_{13}Ce_2Cu_{13}$ phase was evenly distributed on the alloy matrix, which pinned the grain boundary (dispersion strengthening). The addition of Ce resulted in increased amounts of the I-phase. The I-phase area fraction within alloy peaked when the content of Ce was 1%. The phase became finely and uniformly distributed, which significantly elevated alloy hardness [30,31].

When the addition amount of Ce was 2 at%, the amount of the $Al_{13}Ce_2Cu_{13}$ phase increased. Aggregation and growth of the $Al_{13}Ce_2Cu_{13}$ phase were observed, along with decreased inter-atom bonding force and reduced pinning effect on dislocations. The thermal expansion properties and the microhardness of the alloy decreased. As a result, $Al_{13}Ce_2Cu_{13}$ phase had reduced aggregation level whereas mitigated undercooling effect. The amount of the I-phase in the alloy was influenced. The reduction of the I-phase significantly affected the thermal expansion properties and the microhardness of the material.

# 4. Discussion

Varying amounts of Ce significantly affected alloy morphology, microhardness and thermal expansion behaviours. The effects have been discussed in the following section.

## 4.1. The influence of the addition of cerium on the morphology of the alloy

Adding Ce helped to optimize phase morphology present in the alloys. Table 2 shows the degree of mismatch between the solute elements in the Al-Cu-Fe-Ce alloy. It can be seen from the data presented in table 2 that the rare earth Ce has a relatively large atomic radius. The percentages of mismatch between the Al atoms, copper atoms and iron atoms were 27.3%, 42.2% and 43.3%. Ce atoms were almost insoluble in the α-Al crystal lattice. It was difficult to form an alternative solid solution with the Al atoms. The wrapping of the elements around the preferentially nucleated β-phase and I-phase during the solidification process could be attributed to the high surface activity exhibited by the rare earth Ce.

I-phase formation was achieved after the peritectic reaction. During the solidification process, initially, we solidified primary β-phase within the alloy molten at a liquidus temperature. A decrease in the temperature afforded the I-phase. A peritectic reaction along with the β-phase boundaries resulted in the formation of the I-phase. The remaining liquid phase transformed into a quasicrystal approximate phase.

When an appropriate amount of Ce was added to the Al-Cu-Fe alloy, the rare earth Ce surrounding the I-phase formed a package of the phase (figure 4). The growth was inhibited so that a refined I-phase could be formed. The I-phase exhibited the properties of high strength and high hardness [6]. The hardness of the refined I-phase could be significantly increased. The rare earth Ce that wraps the I-phase will adhere to the edge of the I-phase in the form of the $Al_{13}Ce_2Cu_{13}$ phase during the progress of the solidification process. The $Al_{13}Ce_2Cu_{13}$ phase at the edge grew into a long strip along with the grain boundary.

## 4.2. Influence of the addition of cerium on the properties of the alloy

During the arc melting and subsequent suction casting process, the added Ce is at the forefront of grain growth. Ce not only can restrain the growth of the grains but also lead to constitutional supercooling to promote nucleation [32].

Our results suggested that adding Ce at varying amounts resulted in different degrees of component overcooling at the front of AlCuFe alloy solid–liquid interface. We calculated the degree of component undercooling resulting from adding Ce and analysed the effect of changing the component undercooling area on the growth of crystal grains.

First we derived the criterion of component undercooling [33], when the nucleus radius of the primary phase nucleated in the AlCuFe alloy exceeds the critical size of the nucleus. The crystal grains can exist in a stable form and grow. In this study, the temperature distribution in the liquid at the front of the interface is a positive temperature gradient, that is, the crystallization of the liquid phase starts from the region where the fastest cooling occurs, that is, at the lowest temperature.

As the depth of the liquid phase is farther from the solid–liquid interface, the melt temperature becomes higher, so the degree of undercooling of the liquid at the front of the interface decreases with the distance from the interface. Since actual solidification is non-equilibrium solidification, the solute distribution in the liquid phase is not a constant value. When only diffusion occurs in the liquid phase without convection, the composition of the liquid phase at the distance $x$ from the interface can

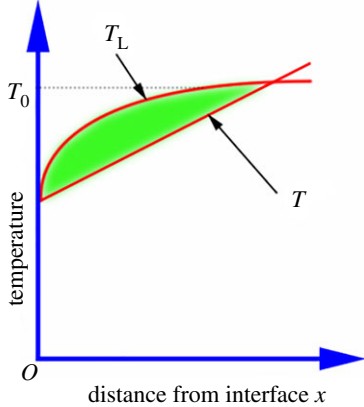

**Figure 11.** Schematic diagram of the formation of supercooling in the front of the alloy solid–liquid interface.

be given as

$$C_L = C_0 \left( 1 + \frac{1 - K_0}{K_0} e^{-Rx/D} \right). \tag{4.1}$$

In formula (4.1), $C_L$ and $C_0$—liquid phase and initial equilibrium components (liquid line is approximately a straight line);

$K_0$—balanced distribution coefficient;

$R$ and $x$—interface movement speed and length from interface;

$D$—diffusion coefficient.

Due to the different compositions of the liquid phase, the theoretical crystallization temperature $T_L$ is also different. Assuming that the liquidus line is straight, the slope is $m$, and the melting point of the pure elements that make up the alloy is $T_A$, the theoretical crystallization temperature of the liquid phase is

$$T_L = T_A - mC_L, \tag{4.2}$$

which is

$$T_L = T_A - mC_0 \left( 1 + \frac{1 - K_0}{K_0} e^{-Rx/D} \right). \tag{4.3}$$

The temperature $T_i$ at the interface ($x = 0$) is

$$T_i = T_A - \frac{mC_0}{K_0}. \tag{4.4}$$

Temperature gradient $G$ can be applied to determine real liquid temperature distribution, and the relationship with the interface distance $x$ is

$$T_D = T_i + Gx, \tag{4.5}$$

$$\text{which is} \quad T_D = T_A - \frac{mC_0}{K_0} + Gx. \tag{4.6}$$

At the real liquid phase temperature lower than the theoretical crystallization temperature, $T_D < T_L$ will result in component overcooling,

$$T_A - \frac{mC_0}{K_0} + Gx < T_A - mC_0 \left( 1 + \frac{1 - K_0}{K_0} e^{-Rx/D} \right), \tag{4.7}$$

$$Gx < \frac{mC_0}{K_0}[(1 - K_0) - (1 - K_0)e^{-Rx/D}] \tag{4.8}$$

$$\text{and} \qquad Gx < \frac{mC_0(1 - K_0)}{K_0}(1 - e^{-Rx/D}). \tag{4.9}$$

According to the differential approximation calculation, when $x$ is very small,

$$e^x = 1 + x. \tag{4.10}$$

**Table 3.** The calculation result of the criterion of component undercooling caused by adding different amounts of Ce.

| the amount of Ce added (wt%) | G/R | $\dfrac{mC_0}{D}\dfrac{(1-K_0)}{K_0}$ |
|---|---|---|
| 0.21 | $2.5\text{–}14.1 \times 10^2$ | $6.38 \times 10^6$ |
| 0.77 | $2.5\text{–}14.1 \times 10^2$ | $2.34 \times 10^7$ |
| 1.3 | $2.5\text{–}14.1 \times 10^2$ | $3.95 \times 10^7$ |
| 1.54 | $2.5\text{–}14.1 \times 10^2$ | $4.68 \times 10^7$ |

We get the following formula:

$$G_x < \frac{mC_0(1-K_0)}{K_0}\frac{R}{D}x. \tag{4.11}$$

Therefore, the conditions for component overcooling are

$$\frac{G}{R} < \frac{mC_0}{D}\frac{(1-K_0)}{K_0}. \tag{4.12}$$

Therefore, during solidification, the solidification rate $R$ increases, the temperature gradient $G$ decreases, the liquid phase slope $m$ increases, the alloy composition $C_0$ increases and the distribution coefficient at equilibrium $K_0$ decreases. It is easy to produce constitutional supercooling. For a certain alloy system, where $m$, $K_0$ and $D$ are fixed values, the conditions that are conducive to the generation of component undercooling include these aspects: the liquid phase has a low temperature gradient, a large solidification rate and a high solute concentration. Figure 11 presents the sketch map for supercooling formation before the interface between liquid and solid.

The addition of Ce determines whether the composition is supercooled before the interface between liquid and solid. In the calculation process, considering earlier studies, $G$ is 4.9–22.5°C cm$^{-1}$, $R$ is 0.016–0.02 cm s$^{-1}$ [34], $m_i = -1.1$, $K_0 = 0.003$, $C_0 = 0.21\text{–}1.54$ wt%, $D = 1.2 \times 10^{-5}$ cm$^2$ s$^{-1}$. In the case where the solute is Ce, the liquidus slope $m_i$ and equilibrium distribution coefficient $k_0$ are determined from the data in the Al-Ce equilibrium phase diagram. The calculation result of the criterion of component undercooling shows that, in solidification, the solidification rate ratio ($G/R$) and liquid phase temperature gradient ranges from 2.5 to $14.1 \times 10^2$. This value is much smaller than the value on the right side of the solute Ce. Therefore, component undercooling occurs during the solidification process, and the component undercooling occurs when the solute Ce is 0.1 wt%. The observation can be attributed to the low equilibrium distribution coefficient (less than 0.1) and diffusion coefficient (approx. $10^{-5}$) of Ce (table 3).

According to the above analysis, as the amount of Ce added increases, the component supercooling zone before the interface between liquid and solid elevates in solidification. The increase in the compositional supercooling zone can activate the nuclei distributed before the interface between liquid and solid to become a new nucleation core. This in turn will increase the nucleation rate of crystal grains and refine the grains.

Ce exerts an important part during grain refinement [24], promoting homogeneous nucleation. An enhancement in the rate of undercooling promotes nucleation. With increased nucleation per unit area, the growth of the nuclei is restricted by the neighbouring crystal nucleus. This results in the formation of a fine alloy microstructure.

A small grain size results in a greater grain boundary amount and more potent force that impedes dislocation movement (fine-grain strengthening). The more the number of grain boundaries, the more hindered is the movement of atoms. The thermal movement of atoms was relatively weaker, along with a stronger inter-atom bonding force. Typically, a lower expansion coefficient was achieved when the atom needs more energy for leaving the equilibrium location.

Fine grains are suggested to potentially contribute to the increase in the strength and material hardness. The grain size dependence of the hardness can be expressed through the classical Hall–Petch relationship [35] as follows:

$$\Delta\sigma = Kd^{-1/2}, \tag{4.13}$$

where

$K$—the Hall–Petch coefficient;

$d$—the average grain size.

According to the Hall–Petch relationship, Al-Cu-Fe-Ce alloy had increased $\Delta\sigma$ compared with Al-Cu-Fe alloy, which was ascribed to the decreasing grain size with an increase in the Ce content. Ce contributes to refining Al-Cu-Fe-Ce alloy grains and improving the hardness of the alloy.

The rare earth Ce can easily combine with Al and Cu to form a white strip or the granular $Al_{13}Ce_2Cu_{13}$ phase. The $Al_{13}Ce_2Cu_{13}$ phase is more likely to accumulate on the grain boundaries, limiting the straightening of the grain boundaries and preventing the grains from growing. According to the Orowan mechanism, when the addition amount of Ce is less than 1 at%, the $Al_{13}Ce_2Cu_{13}$ phase presents a fine and uniform dispersion distribution, pinning dislocations. The significantly improved hardness of the alloy was achieved. However, when the additional amount of Ce is greater than 1 at%, the $Al_{13}Ce_2Cu_{13}$ phase agglomerates, weakening the refining effect of the composition undercooling process.

The alloy hardness is improved, as evidenced by the strengthened solid solution, fine grain and dispersion. The addition of Ce results in the significant refinement of the microstructure of the alloy. I-phase had a great area friction, because the cooling rate was high and Ce atoms were enriched into liquid (before the growing interface) in suction casting. Due to the extremely low Ce solid solubility within the matrix, Ce might probably aggregate at the interface between liquid and solid, giving rise to diffusion layer undercooling, which decreased alloying element diffusion rate while stimulating $Al_{13}Ce_2Cu_{13}$ phase formation across I-phase boundaries. In addition, I-phase growth was observed. Additionally, the suction casting-induced great cooling rate accelerated I-phase nucleation. The grain size was reduced, and the grain boundary areal density remarkably elevated, hindering the dislocation movement and improving the microhardness.

## 5. Conclusion

The Al-Cu-Fe-Ce alloys were prepared following a suction casting process. This study investigated the mechanism by which Ce addition amount affected alloy characteristics and microstructure. Adding an appropriate amount of Ce to Al-Cu-Fe can significantly improve alloy microstructure. Typically, Ce addition significantly affects the grain-refining process. When 1 at% of Ce was added, the coefficient of thermal expansion was reduced by approximately 20% (compared with the alloy devoid of Ce). Under the above conditions, we determined the alloy I-phase area fraction at 63.6%, which was approximately three times higher than that of the alloy devoid of Ce. The hardness of the quaternary Al-Cu-Fe-Ce alloy increased considerably, which was associated with the uniform and fine $Al_{13}Ce_2Cu_{13}$ phase distribution.

Data accessibility. The data are available from the Dryad Digital Repository: https://doi.org/10.5061/dryad.c866t1g6f [36].

The data are provided in the electronic supplementary material [37].

Authors' contributions. J.W. was involved in conceptualization, methodology, formal analysis, investigation, editing and writing original draft. Z.Y. was involved in resources, supervision and data curation. Z.M. was involved in editing. Y.B., H.D. and D.T. were involved in methodology. G.S. was involved in editing. J.L. was involved in validation and supervision.

Competing interest. There are no conflicts to declare.

Funding. This study was financially supported by the Shaanxi Creative Talents Promotion Plan-technological Innovation Team under grant no. 2017KCT-05; Key Project of Equipment Pre-research Field Fund under grant no. 6140922010301; Shaanxi Provincial Key Research and Development Project under grant no. 2019ZDLGY05-09; Shaanxi Provincial Education Department to Serve the Local Special Plan Project under grant no. 19JC022; and Yulin Science and Technology Bureau Project under grant no. 2019-121.

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
