## [Peer Review File · Royal Society Open Science]

Review History

RSOS-210584.R0 (Original submission)

Review form: Reviewer 1

Is the manuscript scientifically sound in its present form?

No

Are the interpretations and conclusions justified by the results?

No

Is the language acceptable?

Yes

Do you have any ethical concerns with this paper?

No

Have you any concerns about statistical analyses in this paper?

Yes

Recommendation?

Major revision is needed (please make suggestions in comments)

Comments to the Author(s)

The authors have reported influence of Ce content on the microstructure and thermal expansion properties of suction cast Al-Cu-Fe alloys. However, the authors should clarify following:

[1] It is not clear how thermal expansion data presented in Fig. 8 is obtained.

[2] There is no definite trend in Fig.8 with change in Ce content. The authors should discuss the trend observed in various samples with Ce content.

[2] The authors should cite the source of equation (1).

Review form: Reviewer 2**Is the manuscript scientifically sound in its present form?**

Yes

Are the interpretations and conclusions justified by the results?

Yes

Is the language acceptable?

Yes

Do you have any ethical concerns with this paper?

No

Have you any concerns about statistical analyses in this paper?

No

Recommendation?

Accept as is

Comments to the Author(s)

The article "Influence of Ce content on the microstructure and thermal expansion properties of suction cast Al-Cu-Fe alloys" is well written in terms of new results and conclusions.

It will be interesting for a wide range of readers. I recommend to publish this article as presented form.

Decision letter (RSOS-210584.R0)

Dear Dr Wang:

Title: Influence of Ce content on the microstructure and thermal expansion properties of suction cast Al-Cu-Fe alloys
Manuscript ID: RSOS-210584

The editor assigned to your manuscript has now received comments from reviewers. We would like you to revise your paper in accordance with the referee and Subject Editor suggestions which can be found below (not including confidential reports to the Editor). Please note this decision does not guarantee eventual acceptance.

Please submit your revised paper before 30-Oct-2021. Please note that the revision deadline will expire at 00.00am on this date. If we do not hear from you within this time then it will be assumed that the paper has been withdrawn. In exceptional circumstances, extensions may be possible if agreed with the Editorial Office in advance. We do not allow multiple rounds of revision so we urge you to make every effort to fully address all of the comments at this stage. If deemed necessary by the Editors, your manuscript will be sent back to one or more of the original reviewers for assessment. If the original reviewers are not available we may invite new reviewers.

Yours sincerely,
Dr Ellis Wilde
Publishing Editor, Journals

On behalf of the Subject Editor Professor Anthony Stace and the Associate Editor Dr Dattatray Late.

RSC Associate Editor
Comments to the Author:
Major Revision

RSC Subject Editor
Comments to the Author:
(There are no comments.)

Reviewers' Comments to Author:

Reviewer: 1

Comments to the Author(s)

The authors have reported influence of Ce content on the microstructure and thermal expansion properties of suction cast Al-Cu-Fe alloys. However, the authors should clarify following:

[1] It is not clear how thermal expansion data presented in Fig. 8 is obtained.

[2] There is no definite trend in Fig.8 with change in Ce content. The authors should discuss the trend observed in various samples with Ce content.

[2] The authors should cite the source of equation (1).

Reviewer: 2

Comments to the Author(s)

The article "Influence of Ce content on the microstructure and thermal expansion properties of suction cast Al-Cu-Fe alloys" is well written in terms of new results and conclusions.

It will be interesting for a wide range of readers. I recommend to publish this article as presented form.

Author's Response to Decision Letter for (RSOS-210584.R0)

See Appendix A.

Decision letter (RSOS-210584.R1)

Dear Dr Wang:

Title: Influence of Ce content on the microstructure and thermal expansion properties of suction cast Al-Cu-Fe alloys

Manuscript ID: RSOS-210584.R1

It is a pleasure to accept your manuscript in its current form for publication in Royal Society Open Science. The chemistry content of Royal Society Open Science is published in collaboration with the Royal Society of Chemistry.

Please see the Royal Society Publishing guidance on how you may share your accepted author manuscript at <https://royalsociety.org/journals/ethics-policies/media-embargo/>. After publication, some additional ways to effectively promote your article can also be found here

<https://royalsociety.org/blog/2020/07/promoting-your-latest-paper-and-tracking-your-results/>.

Yours sincerely,
Dr Ellis Wilde
Publishing Editor, Journals

On behalf of the Subject Editor Professor Anthony Stace and the Associate Editor Dr Dattatray Late.

RSC Associate Editor
Comments to the Author:
Accept as is

Reviewer(s)' Comments to Author:

Appendix A

Dear Dr Ellis Wilde :

Thank you very much for your comments and suggestions. I revised my manuscript item by item according to the comments of the reviewer, and it is marked in red font in the revised version. The following is my response to each reviewer's comments.

Reviewer: 1

Comments to the Author(s)

The authors have reported influence of Ce content on the microstructure and thermal expansion properties of suction cast Al-Cu-Fe alloys. However, the authors should clarify following:

[1] It is not clear how thermal expansion data presented in Fig. 8 is obtained.

Answer: I accept comments from reviewer:1. In the revised manuscript, relevant contents have been added in the experimental details section (Page 5, line 2 to Page 5, line 5 is the added part), the method of obtaining thermal expansion data is explained.

[2] There is no definite trend in Fig.8 with change in Ce content. The authors should discuss the trend observed in various samples with Ce content.

Answer: I accept comments from reviewer:1. In the revised version of the manuscript, section 3.5 adds a discussion on the trend observed in various samples with Ce content (Page 12, line 4 to Page 12, line 8 is the added part).

[3] The authors should cite the source of equation (1).

Answer: I accept comments from reviewer:1. The source of equation (1) is added in the revised manuscript and explained accordingly, and the numbers of other references are readjusted. The new and changed parts are the red font of 5 lines on page 13 and the red font of references from 33 lines on page 23 to 3 lines on page 24. Once again, I would like to thank reviewer :1 for his suggestions.

Reviewer: 2

Comments to the Author(s)

The article "Influence of Ce content on the microstructure and thermal expansion properties of suction cast Al-Cu-Fe alloys" is well written in terms of new results and conclusions.

It will be interesting for a wide range of readers. I recommend to publish this article as presented form.

Answer: Thank you very much for Reviewer: 2's comments on this manuscript.

In addition,

1. In the author column, the deleted authors are Zhen Wang and Jiachen Zhang, and

the added authors are Zhijun Ma, Yaping Bai, Hongbo Duan and Dong Tao. The change of the author has been approved by all other authors.

2. In terms of the format of the manuscript, change to center alignment for Figure 4 and references.

Your opinions and suggestions really helped me a lot. I put a lot of effort into this review. I hope you are satisfied.

Thank you again for your time and patience. Looking forward to your reply.

Yours Sincerely

Juan Wang